# Comparison of Amsler–Krumeich and Sandali Classifications for Staging Eyes with Keratoconus

Giuseppe Giannaccare [1,*][iD], Gianluca Murano [1], Adriano Carnevali [1], Angeli Christy Yu [2], Sabrina Vaccaro [1], Gianfranco Scuteri [1][iD], Laura Maltese [1] and Vincenzo Scorcia [1,*][iD]

1 Department of Ophthalmology, University Magna Graecia of Catanzaro, 88100 Catanzaro, Italy; gianlucamurano89@gmail.com (G.M.); adrianocarnevali@unicz.it (A.C.); sabrina_vaccaro@libero.it (S.V.); gianfrancoscuteri@gmail.com (G.S.); laura.maltese@studenti.unicz.it (L.M.)
2 Department of Translational Medicine, University of Ferrara, 44121 Ferrara, Italy; angelichristy.yu@unife.it
* Correspondence: giuseppe.giannaccare@unicz.it (G.G.); vscorcia@libero.it (V.S.); Tel.: +39-3317186201 (G.G.); +39-3334800929 (V.S.)

**Abstract:** Keratoconus (KC) is the most common corneal ectasia characterized by progressive corneal thinning, protrusion, and irregular astigmatism. The Amsler–Krumeich classification based on the analysis of corneal topography, corneal thickness, refraction and biomicroscopy is the most commonly used; recently, a new classification based on anterior segment Optical Coherence Tomography was introduced by Sandali and colleagues. Since there is no information about the possible agreement between these two classifications, the aim of this study is to compare the stratification of consecutive KC patients using the Amsler–Krumeich and Sandali classifications, and to further ascertain KC cases in which one classification is preferred over the other. Overall, 252 eyes of 137 patients ($41.45 \pm 16.93$ years) were analyzed: in 156 eyes (61.9%), the Amsler and Sandali staging differed in one stage while in 75 cases (29.8%) it differed in two or more stages. In 222 eyes (88.1%), the Sandali staging was higher compared to the Amsler one. These results show that the two classifications are not fully interchangeable: the Amsler–Krumeich classification is more appropriate in identifying and longitudinally monitoring patients with early stages of KC, while the Sandali classification for the diagnosis and follow-up of patients with more advanced stages, particularly when a surgical planning has to be chosen.

**Keywords:** keratoconus; cornea; Amsler–Krumeich classification; Sandali classification; anterior segment optical coherence tomography (AS-OCT)



## 1. Introduction

Keratoconus (KC) is a corneal disorder characterized by ectasia and thinning of the cornea, which often causes high and irregular myopic astigmatism [1]. Therefore, KC patients experience suboptimal to poor vision despite wearing spectacles or contact lenses, especially if corneal scars or elevated irregular astigmatism are present. The diagnosis and classification of the disease have evolved in last decades thanks to the continuous advances in technology [2]. The initial classifications were based on the detection of irregular corneal astigmatism by means of a placid disc or the Javal ophthalmometer, along with the presence of characteristic signs in the biomicroscopic examination of the cornea [3]. Placido disc-based corneal topography is a sensitive and specific diagnostic tool that examines the anterior surface of the cornea but not the posterior corneal curvature, which is crucial in the early detection of KC. With the advent of the rotating Scheimpflug imaging and slit-scanning topography, anterior and posterior elevation measurements as well as curvature analysis have become available in the clinical setting [4,5]. With the height data, the anterior protrusion and the corneal shape parameter changes can be assessed, since they are different from the curvature map assessments of the relative distortions of the cornea, and can provide useful diagnostic information for the detection of KC [6].

The Amsler–Krumeich classification is currently still one of the most widely used system for identification of keratoconus and assessment of disease progression. This grading system is based on topographic analysis of the anterior corneal surface, corneal thickness, refraction and biomicroscopy [7]. However, this system does not consider the morphologic changes in the corneal microarchitecture of keratoconus eyes.

Thanks to the use of anterior segment Optical Coherence Tomography (AS-OCT), a non-contact technique based on the principles of low-coherence interferometry [8,9], Sandali and coauthors described a new classification based on structural corneal changes occurring in keratoconus throughout the evolution of the disease [10]. Based on the structural changes that occur in the corneal microarchitecture of keratoconus eyes, this system also allows grading of the disease severity.

To date, there is no information about the possible agreement between these two different KC classifications. Therefore, the aim of this study is to compare the stratification of consecutive KC patients according to disease severity using the Amsler–Kumeich and Sandali classifications, and to further ascertain KC cases in which the use of one classification is preferred over the other one.

## 2. Materials and Methods

The study followed the tenets of the 2013 Declaration of Helsinki and was approved by the local Ethics Committee (Comitato Etico Regione Calabria). Consecutive patients affected by KC who attended the cornea clinic for a routine examination from September 2015 to November 2020 were enrolled. Informed consent was obtained from all patients before data analysis.

Keratoconus was clinically defined as the presence of signs including stromal thinning, Vogt striae, Fleischer rings, and topographic findings consistent with keratoconus. Eyes with a history of keratoplasty were excluded. For patients who underwent more than one visit during the study period, the first examination was used for statistical purposes.

Each patient underwent complete ophthalmological examination including visual acuity testing and slit lamp examination; a Casia 1 (Tomey Corp., Nagoya, Japan) was used to perform topographical maps for the calculation of keratometric (K mean) and pachymetric values as well as AS-OCT scans for the detection of corneal layer abnormalities. Data were collected by the same examiner (GG) and also independently analyzed by another operator (GM). All keratoconic eyes of patients enrolled were classified according to the Amsler–Krumeich and Sandali classifications (Table 1).

**Table 1.** Comparison of the Amsler–Krumeich and Sandali classifications.

| Amsler–Krumeich | Sandali |
|---|---|
| **STAGE I:**<br>Eccentric steepening myopia/astigmatism < 5.00 D<br>Mean keratometry value < 48.0 D | **STAGE 1:**<br>Thinning of epithelial and stromal layers at the conus<br>Corneal layers with normal aspect |
| **STAGE II:**<br>Myopia/astigmatism > 5.00 D but <8.00 D<br>Mean keratometry value < 53.0 D<br>Absence of scarring<br>Minimal apical corneal thickness > 400 μm | **STAGE 2A:**<br>Hyperreflective anomalies at the Bowman's layer level with epithelial thickening |
| | **STAGE 2B:**<br>Hyperreflective anomalies at the Bowman's layer level with epithelial thickening plus stromal opacities |
| **STAGE III:**<br>Myopia/astigmatism > 8.00 D but <10.00 D<br>Mean keratometry value > 53.0 D<br>Absence of scarring<br>Minimal apical corneal thickness < 400 μm but >300 μm | **STAGE 3A:**<br>Posterior displacement of the hyperreflective structures at the Bowman's layer level with increased epithelial thickening<br>Stromal thinning |
| | **STAGE 3B:**<br>Posterior displacement of the hyperreflective structures at the Bowman's layer level with increased epithelial thickening<br>Stromal thinning plus the presence of stromal opacities |
| **STAGE IV:**<br>Refraction not possible<br>Mean keratometry value > 55.0 D<br>Central corneal scarring<br>Minimal apical corneal thickness < 300 μm | **STAGE 4:**<br>Pan stromal scar |
| | **STAGE 5A:**<br>Acute onset with Descemet's membrane rupture with dilacerations of collagen lamellae, large fluid filled intrastromal cysts and epithelial edema formation |
| | **STAGE 5B:**<br>Healing stage of 5A with panstromal scarring with remaining aspect of Descemet's membrane rupture. |

All data collected from the study were entered into an electronic database using the Excel 2007 software (Microsoft, Redmond, Washington, DC, USA). Data were expressed as mean ± standard deviation (SD) for continuous variables and as individual counts and percentages for categorical variables. The percentage of eyes belonging to each stage of the two classifications as well as the percentage of patients for whom the classifications differ for one or more stages were calculated. When comparing the two classification systems, for statistical purposes, the Sandali's stages 2A–B were considered stage 2, the Sandali's stages 3A–B were considered stage 3, and the Sandali's stages 4, 5A–B were considered stage 4. The percentage of patients undergoing keratoplasty for each stage of both classifications were also calculated.

## 3. Results

Overall, 252 eyes of 137 consecutive patients with KC (86 males, 51 females; 41.45 ± 16.93 years) were included. According to the Amsler–Krumeich classification, the studied eyes were divided as follows: stage I (n = 3 eyes, 1.2%); stage II (n = 144 eyes, 57.1%), stage III (n 69 eyes, 27.4%), and stage IV (n = 36 eyes, 14.3%). According to the Sandali classification, the studied eyes are divided as follows: stage 1 (n = 206 eyes, 81.7%), stage 2 (n = 16 eyes, 6.4%) which was subdivided into stage 2A (n = 9 eyes, 3.6%) and stage 2B (n = 7, 2.8%), stage 3 (n = 2 eyes, 0.8%) of which one in stage 3A (0.4%) and one in stage 3B (0.4%), stage 4 (n = 25 eyes, 9.9%), and stage 5 (n = 3 eyes 1.2%) of which one eye in stage 5A (0.4%) and another eye in stage 5B (0.4%). Tables 2 and 3 report the clinical parameters of each stage of the Amsler–Krumeich and Sandali classifications, respectively.

**Table 2.** Mean keratometry, cylinder and pachymetry values in the group classified with the Amsler–Krumeich classification.

| | All Eyes (n = 252) | Stage 1 (n = 3) | Stage 2 (n = 144) | Stage 3 (n = 69) | Stage 4 (n = 36) |
|---|---|---|---|---|---|
| K steep, anterior (D) mean ± SD | 59.22 ± 8.55 | 49.40 ± 1.27 | 54.76 ± 3.66 | 61.14 ± 5.82 | 74.21 ± 8.68 |
| K flat, anterior (D) mean ± SD | 55.34 ± 7.45 | 47.53 ± 0.75 | 51.45 ± 2.99 | 57.03 ± 5.09 | 68.34 ± 8.03 |
| Cyl, anterior (D) mean ± SD | 3.87 ± 2.39 | 1.80 ± 0.96 | 3.31 ± 1.77 | 4.10 ± 2.44 | 5.86 ± 3.27 |
| K steep, posterior (D) mean ± SD | −8.20 ± 1.44 | −6.43 ± 0.25 | −7.42 ± 0.71 | −8.59 ± 0.93 | −10.75 ± 1.19 |
| K flat, posterior (D) mean ± SD | −7.52 ± 1.28 | −6.03 ± 0.32 | −6.83 ± 0.65 | −7.87 ± 0.91 | −9.69 ± 1.11 |
| Cyl, posterior (D) mean ± SD | 0.68 ± 0.38 | 0.40 ± 0.10 | 0.58 ± 0.27 | 0.69 ± 0.33 | 1.04 ± 0.57 |
| TCT (μm) mean ± SD | 406.14 ± 72.35 | 523.67 ± 11.93 | 450.94 ± 27.39 | 374.83 ± 27.33 | 277.19 ± 67.35 |

Cyl = cylinder; K = keratometry value; SD = standard deviation; TCT = thinnest corneal thickness.

**Table 3.** Mean keratometry, cylinder and pachymetry values in the group classified with the Sandali classification.

| | All Eyes (n = 252) | Stage 1 (n = 206) | Stage 2 (n = 16) | Stage 3 (n = 2) | Stage 4 (n = 25) | Stage 5 (n = 3) |
|---|---|---|---|---|---|---|
| K steep, anterior (D) mean ± SD | 59.22 ± 8.55 | 56.40 ± 5.03 | 75.14 ± 10.71 | 69.55 ± 8.83 | 69.08 ± 8.75 | 78.97 ± 2.45 |
| K flat, anterior (D) mean ± SD | 55.34 ± 7.45 | 52.89 ± 4.32 | 69.71 ± 9.94 | 63.85 ± 6.71 | 63.86 ± 7.57 | 70.60 ± 2.36 |
| Cyl, anterior (D) mean ± SD | 3.87 ± 2.39 | 3.51 ± 1.97 | 5.43 ± 2.47 | 5.70 ± 2.12 | 5.20 ± 3.70 | 8.47 ± 4.30 |
| K steep, posterior (D) mean ± SD | −8.20 ± 1.44 | −7.71 ± 0.92 | −10.56 ± 1.45 | −10.15 ± 1.90 | −10.19 ± 1.27 | −11.20 ± 1.67 |
| K flat, posterior (D) mean ± SD | −7.52 ± 1.28 | −7.10 ± 0.85 | −9.62 ± 1.20 | −9.35 ± 2.05 | −9.26 ± 1.18 | −9.47 ± 1.28 |
| Cyl, posterior (D) mean ± SD | 0.68 ± 0.38 | 0.61 ± 0.29 | 0.94 ± 0.51 | 0.80 ± 0.14 | 0.90 ± 0.53 | 1.70 ± 0.98 |
| TCT (μm) mean ± SD | 406.14 ± 72.35 | 429.17 ± 47.62 | 306.75 ± 67.52 | 286.00 ± 83.43 | 300.44 ± 81.76 | 315.67 ± 80.85 |

Cyl = cylinder; K = keratometry value; SD = standard deviation; TCT = thinnest corneal thickness.

In 156 eyes (61.9% of total), the Amsler and Sandali staging systems differed in one stage, in 75 cases (29.8%) they differed in two or more stages. In 222 eyes (88.1%), the Sandali staging was higher compared to the Amsler one; in 21 eyes (8.3%), the two staging systems provided the same value, while only in nine cases (3.6%) the Amsler staging provided a higher grade compared with the Sandali system.

Of 252 eyes, 101 eventually underwent keratoplasty for visual rehabilitation during the study period. The mean time from diagnosis to keratoplasty was $16.74 \pm 16.7$ months (1–69 months). Of these, 96 eyes underwent deep anterior lamellar keratoplasty (DALK), while 5 underwent mushroom-shaped penetrating keratoplasty (PK). According to the Amsler–Krumeich classification, the eyes undergoing keratoplasty were scored as follows: stage I (n = 0), stage II (n = 29 eyes, 28.7% of the overall surgical cases), stage III (n = 44 eyes, 43.6%), stage IV (n = 28 eyes, 27.7%), while according to the Sandali classification, they were scored as follows: stage 1 (n = 64 eyes, 63.5% of the overall surgical cases); stage 2 (n = 11 eyes, 10.9%); stage 3 (n = 2 eyes, 1.9%), stage 4 (n = 22 eyes, 21.8%), and stage 5 (n = 2 eyes, 1.9%).

Two representative cases of KC patients in which the two classifications were not consistent are shown in Figures 1 and 2.

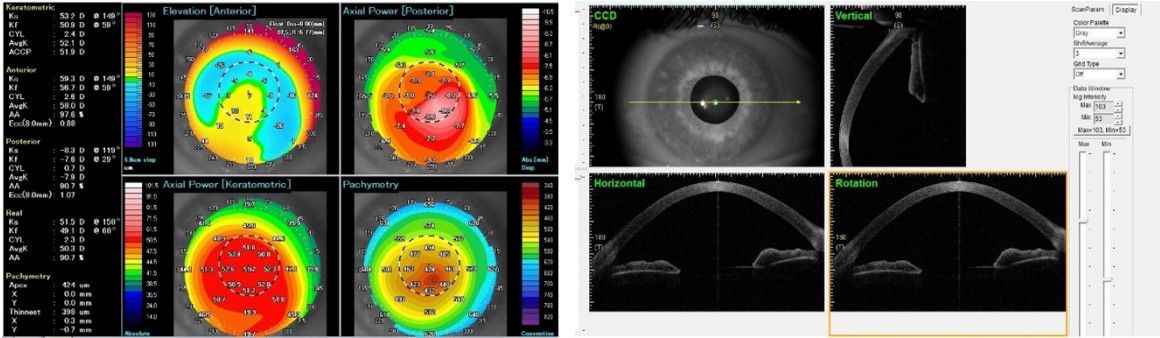

**Figure 1.** Representative case of a patient with keratoconus in which the two classifications were not consistent (the Amsler–Krumeich score higher than the Sandali one). The disease was classified as stage III according to the Amsler–Krumeich classification while as stage 1 according to the Sandali classification.

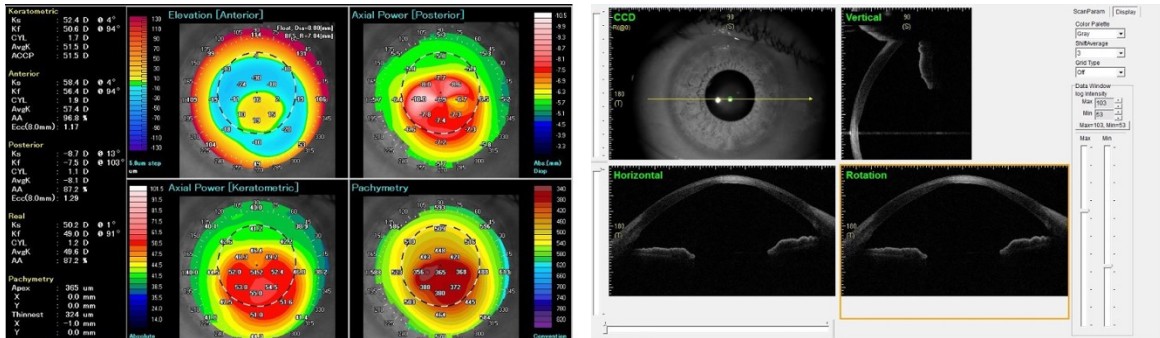

**Figure 2.** Representative case of a patient with keratoconus in which the two classifications were not consistent (the Sandali score higher than the Asmler–Krumeich one). The disease was classified as stage I according to the Amsler–Krumeich classification while as stage 4 according to the Sandali classification.

## 4. Discussion

Nowadays, different types of KC classifications are available, depending on the parameters included in the analysis. Among these, the Amsler–Krumeich and Sandali classifications based, respectively, on the corneal map and AS-OCT findings are the two most commonly used. However, to date, no direct comparison between these two classifications in the same group of KC patients has been performed.

In the present study, we pointed out the different staging of KC obtained according to the two classifications, showing the lack of agreement between the curvature alterations observed with the corneal topography and structural changes detected by means of AS-OCT.

The Amsler–Krumeich classification is useful in the detection of keratoconus and monitoring of its progression. This is particularly important in screening candidates for refractive corneal surgery or lens-based procedures using premium IOLs such as multifocal IOLs. Since only the anterior corneal curvature is considered in the Amsler–Krumeich classification, several other classification systems have been proposed. Using a Scheimpflug system, the ABCD grading system by Belin and associates incorporates the anterior and posterior corneal curvature, the thinnest pachymetry value, and the vision to characterize the tomographic and functional aspects of keratoconic corneas [11]. As tomography provides information regarding the subtle changes in the posterior corneal curvature, which precedes the anterior steepening, this tomography-based classification system has been proposed to allow early diagnosis of subclinical disease. It does not, however, characterize the histopathological structural changes that result in the topographic alterations associated with keratoconus.

High-resolution OCT provides evaluation of each corneal layer and allows assessment of the corneal structural changes in keratoconus. The OCT-based classification proposed by Sandali et al. establishes a grading system based on the changes observed in progressive corneal ectasia. In advanced cases of keratoconus, wherein the repeatability of corneal topography is less reliable, the OCT allows continued monitoring of the structural alterations [10].

In patients undergoing keratoplasty for severe KC with the presence of stromal scars, the AS-OCT analysis is crucial for the choice of the proper surgical planning [12–14]. Borderie and co-workers analyzed predictive factors for the formation and type of a big bubble (BB) during DALK and found that type 1 BB formation was significantly associated with the absence of scars in the posterior stroma (stages 1–3 of the Sandali classification); conversely, the presence of posterior scars represents a significant risk factor for type 2 BB formation [15]. In fact, it has been hypothesized that deep scarring may fuse the PDL to the posterior stroma or degrade the PDL itself, preventing type 1 BB formation owing to the direct air dispersion at the level of DM. This poor prognostic value of posterior scars was found to be even stronger in patients with KC compared to patients with other corneal diseases [16,17].

Since the floor of a type 2 BB consists only of DM, it carries a high risk of perforation, which can mandate a conversion to PK. Furthermore, even in the absence of a frank perforation, the occurrence of a type 2 BB has been associated with an increased risk of postoperative double anterior chamber formation requiring further surgical interventions. As such, the formation of type 2 BB increases the intraoperative challenges of DALK surgery. For this reason, a detailed preoperative assessment of KC by means of AS-OCT and its staging, according to the Sandali classification, may provide useful information for the surgical planning of DALK in terms of both the timing of surgery and the technique employed. Surgical intervention before the development of deep stromal scarring associated in BB-DALK with subsequent type 2 bubble formation and increased risk of conversion to PK is desirable. On the other hand, in severe KC cases, when posterior/panstromal scars are present, the use of manual techniques should be taken into account in order to avoid intraoperative complications and reducing the risk of PK conversion.

We recognized that, despite the two classifications are the most used ones, they suffer from some limitations, as stated by the Global Consensus on Keratoconus [18]. Indeed, the panel agreed that a suitable classification system using this additional information currently does not exist and that further studies that correlate clinical findings, such as visual performance with corneal topometric and tomographic parameters, are needed. Furthermore, the number of stages of the two classifications differs (4 stages for Amsler–Krumeich and 5 stages for Sandali) and they were readapted for statistical purposes.

## 5. Conclusions

In conclusion, the Amsler–Krumeich and Sandali classifications are useful in patients with KC; however, they are not fully interchangeable since they detect different alterations. In particular, the use of the Amsler–Krumeich classification is more appropriate for identifying patients with early stages of the disease and for their longitudinal monitoring. Conversely, the Sandali classification, based on the analysis of the images obtained with AS-OCT, is useful for the diagnosis and follow-up of patients with advanced stages of KC, particularly when a surgical planning has to be chosen.

**Author Contributions:** Conceptualization, G.G. and V.S.; methodology, G.G, G.M., S.V., L.M., A.C.Y., G.S.; formal analysis, G.G. and V.S.; resources, G.G., A.C., V.S.; data curation, G.M., A.C.Y., S.V., L.M., G.S.; writing—original draft preparation, G.G., G.M., A.C., A.C.Y., S.V., L.M., G.S.; writing—review and editing, G.G., V.S.; supervision, V.S.; project administration, V.S.; funding acquisition, V.S. All authors have read and agreed to the published version of the manuscript.

**Funding:** This review received no external funding.

**Institutional Review Board Statement:** The study followed the tenets of the 2013 Declaration of Helsinki and was approved by the local Ethics Committee (Comitato Etico Regione Calabria, protocol n. 12-2021, date of approval 21 jan 2021).

**Informed Consent Statement:** Informed consent was obtained from all subjects involved in the study.

**Data Availability Statement:** Data available on request due to restrictions (privacy).

**Conflicts of Interest:** The authors declare no conflict of interest.

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
