# Peer review of "Comparison of Amsler–Krumeich and Sandali Classifications for Staging Eyes with Keratoconus"

_applsci, doi:10.3390/app11094007_

Round 1
Reviewer 1 Report
The authors have compared the Amsler Krumeich and Sandali Classifications for classify eyes with keratoconus. The article is well written but does not have enough clinical relevance. Actually, in accordance with Global Consensus on Keratoconus (2015 Cornea) the historical Amsler–Krumeich classification fails to address current information and technological advances and the Sandali Classification is not commonly used in clinical practice.
In addition, the Amsler classification is divided into 4 degrees of severity, while the Sandali classification has 5, so they cannot be comparable.
The authors should describe each classification in a table to facilitate the reading of the article.
It is necessary to include the mean values of K and pachymetry, both in the groups using Amsler and Sandali classification. It would have been interesting to calculate cutoff values with K and pachymetry using ROC curves in the cases classified with Sandali.
Finally, the authors have included a large majority of eyes with keratoplasty, which represents a bias in the analysis. Virgin keratoconus eyes must be used in these types of studies.
Author Response
- The authors have compared the Amsler Krumeich and Sandali Classifications for classify eyes with keratoconus. The article is well written but does not have enough clinical relevance. Actually, in accordance with Global Consensus on Keratoconus (2015 Cornea) the historical Amsler–Krumeich classification fails to address current information and technological advances and the Sandali Classification is not commonly used in clinical practice. In addition, the Amsler classification is divided into 4 degrees of severity, while the Sandali classification has 5, so they cannot be comparable.
Response: We added a limitation statement in the discussion section (Page 7-8, Line 252-283) as follows: “We recognized that despite the two classifications are the most used ones, they suffer from some limitations, as stated by the Global Consensus on Keratoconus [18]. Indeed, the panel agreed that a suitable classification system using this additional information currently does not exist and that further studies that correlate clinical findings such as visual performance with corneal topometric and tomographic parameters are needed. Furthermore, the number of stages of the 2 classifications differs (4 stages for Amsler-Krumeich and 5 stages for Sandali) and they were readapted for statistical purposes.”
- The authors should describe each classification in a table to facilitate the reading of the article.
Response: We have added the Amsler-Krumeich and OCT classification system by Sandali et al. in Table 1.
- It is necessary to include the mean values of K and pachymetry, both in the groups using Amsler and Sandali classification. It would have been interesting to calculate cutoff values with K and pachymetry using ROC curves in the cases classified with Sandali.
Response: We have added the mean keratometry and pachymetry values in both groups as Table 2 and 3 (Page 4-5). The main purpose of the study was to compare 2 classification systems for keratoconus. We did not evaluate normal controls, thus, ROC curve analysis to obtained cutoff of Sandali OCT classification could not be performed.
- Finally, the authors have included a large majority of eyes with keratoplasty, which represents a bias in the analysis. Virgin keratoconus eyes must be used in these types of studies.
Response: We have clarified this issue in the Results section (Page 4, Line 125-127). In fact, only virgin keratoconus eyes were included in this study. In order to clarify this issue, we added for patients who underwent keratoplasty the time interval (mean ± SD) between study visit and surgery.
“Of 252 eyes, 101 eventually underwent keratoplasty for visual rehabilitation during the study period. Mean time from diagnosis to keratoplasty was 16.74 ± 16.7 months (1-69 months).”
Reviewer 2 Report
The treatment of ecstatic diseases have advanced dramatically in the last decade, however, the most widely used Amsler-Krumeich grading system for keratoconus is over 70 years old and has some limitations. In this report, Dr. Giannaccare etc compared the clinical validity of Amsler Krumeich and Sandali Classifications, summarized the stratification of consecutive KC patients according to disease severity using the two classifications. The authors concluded that the Amsler-Krumeich classification is more accurate in identifying patients with early stages of the disease and for their longitudinal monitoring. Conversely, Sandali classification is useful for the diagnosis and follow-up of patients with advanced stages of KC, particularly when a surgical planning has to be chosen.
The manuscript is written concisely and is very thorough, explains everything very carefully. I could not find any major deficits in the manuscript.
Author Response
1. The treatment of ecstatic diseases have advanced dramatically in the last decade, however, the most widely used Amsler-Krumeich grading system for keratoconus is over 70 years old and has some limitations. In this report, Dr. Giannaccare etc compared the clinical validity of Amsler Krumeich and Sandali Classifications, summarized the stratification of consecutive KC patients according to disease severity using the two classifications. The authors concluded that the Amsler-Krumeich classification is more accurate in identifying patients with early stages of the disease and for their longitudinal monitoring. Conversely, Sandali classification is useful for the diagnosis and follow-up of patients with advanced stages of KC, particularly when a surgical planning has to be chosen.
The manuscript is written concisely and is very thorough, explains everything very carefully. I could not find any major deficits in the manuscript.
Response: We thank the reviewer for his appreciation of our work.
Reviewer 3 Report
Thank you for consulting me to review this interesting manuscript. In this study, the authors compare the Amsler-Krumeich classification of keratoconus with a novel method proposed by Sandali and colleagues using AS-OCT. I have the following feedback for the authors:
- There should be increased background information within the Introduction on the two classification methods. They should be outlined fully within a table or a figure to show the different stages and corresponding clinical presentation/parameters. This should be removed from the Materials and Methods entirely.
- Given that the Materials and Methods will be significantly shortened per the above comment, the authors should expand within the Materials and Methods on their protocol. They should detail inclusion criteria for patients, the number of clinicians involved in examination, how data was collected and where it was stored, and who performed the classifications. Details of the statistical analyses performed should also be provided.
- The authors claim that the Amsler-Krumeich classification is more accurate for mild KC while the Sandali method is more applicable to late-stage disease. How did they reach this conclusion? What makes Amsler-Krumeich better for early-stage disease? What makes Sandali better for late-stage disease? The statistical data from the authors’ study does not provide the evidence to substantiate these claims. The authors should remove this from their study entirely as it is misleading.
- The authors spend considerable time in their Discussion focussing on the utility of AS-OCT in operative planning. This is helpful, however, there should also be discussion of the utility of the Amsler-Krumeich method.
- The authors do not comment on any of the many other classification methods for keratoconus that exist. For instance, the ABCD criteria offers an alternative to Amsler-Krumeich as well. It would be interesting to see how the authors’ patient population scores with different keratoconus staging methods.
Author Response
1. Thank you for consulting me to review this interesting manuscript. In this study, the authors compare the Amsler-Krumeich classification of keratoconus with a novel method proposed by Sandali and colleagues using AS-OCT. I have the following feedback for the authors:
There should be increased background information within the Introduction on the two classification methods. They should be outlined fully within a table or a figure to show the different stages and corresponding clinical presentation/parameters. This should be removed from the Materials and Methods entirely.
Reply: We have added background information regarding both classification methods in the Introduction (Page 2, Line 55-65). We have outlined the Amsler-Krumeich and Sandali OCT classification system as Table 1.
“The Amsler-Krumeich classification is still currently one of the most widely used for system for identification of keratoconus and assessment of disease progression. This grading system is based on topographic analysis of the anterior corneal surface, corneal thickness, refraction and biomicroscopy [7]. However, this system does not consider the morphologic changes in the corneal microarchitecture of keratoconus eyes.
Thanks to the use of anterior segment Optical Coherence Tomography (AS-OCT), a non-contact technique based on the principles of low-coherence interferometry [8,9], Sandali and coauthors described a new classification based on structural corneal changes occurring in keratoconus throughout the evolution of the disease [10]. Based on the structural changes that occur in the corneal microarchitecture of keratoconus eyes, this system also allows grading of the disease severity.”
2. Given that the Materials and Methods will be significantly shortened per the above comment, the authors should expand within the Materials and Methods on their protocol. They should detail inclusion criteria for patients, the number of clinicians involved in examination, how data was collected and where it was stored, and who performed the classifications. Details of the statistical analyses performed should also be provided.
Reply: We have revised the Methods which includes details on inclusion criteria, data collection and analysis (Page 2-3, Line 78-105).
“Keratoconus was clinically defined as the presence of signs including stromal thinning, Vogt stria, Fleischer ring, topographic findings consistent with keratoconus. Eyes with a history of keratoplasty were excluded. For patients who underwent more than 1 visit during the study period, the first examination was used for statistical purposes.
Each patient underwent complete ophthalmological examination including visual acuity testing and slit lamp examination; Casia 1 (Tomey Corp., Nagoya, Japan) was used to perform topographical maps for the calculation of keratometric (K mean) and pach-ymetric values as well as for AS-OCT scans for the detection of corneal layer abnormalities. Data were collected by the same examiner (GG) and also independently analyzed by another operator (GM). All keratoconic eyes of patients enrolled were classified according to Amsler-Krumeich and Sandali classifications (Table 1).
All data collected from the study were entered into an electronic database using Excel 2007 software (Microsoft, Redmond, Washington). Data were expressed as mean ± standard deviation (SD) for continuous variables and as individual counts and per-centages for categorical variables. The percentage of eyes belonging to each stage of the 2 classifications as well as the percentage of patients for whom the classifications differ for 1 or more stages were calculated. When comparing the two classification systems, for sta-tistical purposes Sandali’s stages 2A-B were considered stage 2, Sandali’s stages 3A-B were considered stage 3, Sandali’s stages 4, 5A-B were considered stage 4. The percentage of patients undergoing keratoplasty for each stage of both classifications were also calculated.”
3. The authors claim that the Amsler-Krumeich classification is more accurate for mild KC while the Sandali method is more applicable to late-stage disease. How did they reach this conclusion? What makes Amsler-Krumeich better for early-stage disease? What makes Sandali better for late-stage disease? The statistical data from the authors’ study does not provide the evidence to substantiate these claims. The authors should remove this from their study entirely as it is misleading.
Reply: We apologize for the confusion. We have revised the Discussion section (Page 7, Line 209-227).
“Amsler-Krumeich classification is useful in the detection of keratoconus and moni-toring of its progression. This is particularly important in screening candidates for re-fractive corneal surgery or lens-based procedures using premium IOLs such as multifocal IOLs. Since only the anterior corneal curvature is considered in the Amsler-Krumeich classification, several other classification systems have been proposed. Using a Scheimpflug system, the ABCD grading system by Belin and associates incorporates the anterior and posterior corneal curvature, thinnest pachymetry value, and vision to characterize the tomographic and functional aspects of keratoconic corneas [11]. As to-mography provides information regarding the subtle changes in the posterior corneal curvature, which precedes anterior steepening, this tomography-based classification system has been proposed to allow early diagnosis of subclinical disease. It does not however, characterize the histopathological structural changes that result in the topo-graphic alterations associated with keratoconus.
High- resolution OCT provides evaluation of each corneal layer and allows as-sessment of the corneal structural changes in keratoconus. The OCT-based classification proposed by Sandali et al establishes a grading system based on the changes observed in progressive corneal ectasia. In advanced cases of keratoconus wherein the repeatability of corneal topography is less reliable, the OCT allows continued monitoring of the structural alterations [10].”
4. The authors spend considerable time in their Discussion focussing on the utility of AS-OCT in operative planning. This is helpful, however, there should also be discussion of the utility of the Amsler-Krumeich method.
Reply: We have discussed further the Amsler-Krumeich classification in the Discussion (Page 7, Line 209-212).
“Amsler-Krumeich classification is useful in the detection of keratoconus and monitoring of its progression. This is particularly important in screening candidates for refractive corneal surgery or lens-based procedures using premium IOLs such as multifocal IOLs.”
5. The authors do not comment on any of the many other classification methods for keratoconus that exist. For instance, the ABCD criteria offers an alternative to Amsler-Krumeich as well. It would be interesting to see how the authors’ patient population scores with different keratoconus staging methods.
Reply: We have added a discussion on the ABCD criteria (Page 7, Line 212-221).
“Since only the anterior corneal curvature is considered in the Amsler-Krumeich classification, several other classification systems have been proposed. Using a Scheimpflug system, the ABCD grading system by Belin and associates incorporates the anterior and posterior corneal curvature, thinnest pachymetry value, and vision to characterize the tomographic and functional aspects of keratoconic corneas [11]. As tomography provides information regarding the subtle changes in the posterior corneal curvature, which precedes anterior steepening, this tomography-based classification system has been proposed to allow early diagnosis of subclinical disease. It does not however, characterize the histopathological structural changes that result in the topographic alterations associated with keratoconus.”
Round 2
Reviewer 1 Report
The authors have made a great effort to improve the article.
Author Response
Thanks for your positive comments.
Reviewer 3 Report
The authors have substantially improved their manuscript and addressed my concerns thoroughly, however, they still claim within their Abstract that 'Amsler-Krumeich classification should be preferred in earlier stages of KC thanks to the higher diagnostic performance of milder alterations, while Sandali classification should be adopted in advanced stages in order to set up the proper surgical planning." This statement is incorrect and unsupported by the data. As mentioned in the previous set of reviews, the authors should remove this wording. They have done so from the Discussion, but need to re-read the manuscript to ensure this scientific message is removed throughout. There is no claim from the manuscript to support this assertion.
Author Response
Thanks for your comments. We agree with you that the same modifications done in the full text should be reported also in the abstract text. Therefore, we modified the abstract's conclusions as follows:
"This results show that the 2 classifications are not fully interchangeable: the Amsler-Krumeich classification is more appropriate in identifying and longitudinally monitoring patients with early stages of KC, while Sandali classification for the diagnosis and follow-up of patients with more advanced stages, particularly when a surgical planning has to be chosen."